# Evolution of Malaria Incidence in Five Health Districts, in the Context of the Scaling up of Seasonal Malaria Chemoprevention, 2016 to 2018, in Mali

**DOI:** 10.3390/ijerph18020840

**Published:** 2021-01-19

**Authors:** Aissata Sacko, Issaka Sagara, Ibrahima Berthé, Modibo Diarra, Mady Cissoko, Souleymane S. Diarra, Drissa Coulibaly, Moussa Sanogo, Alassane Dicko

**Affiliations:** 1Department of Public Health, Faculty of Medicine and Odonto-Stomatology, University of Sciences, Techniques and Technology of Bamako, Bamako BP 1805, Mali; moussanogo2002@yahoo.fr (A.S.); diarrasoul24@yahoo.fr (S.S.D.); aissata1905@yahoo.fr (M.S.); adicko@icermali.org (A.D.); 2Malaria Research and Training Center, Department of Epidemiology of Parasitic Diseases, Faculty of Pharmacy, University of Sciences Techniques and Technology of Bamako, Bamako BP 1805, Mali; modibod@incermali.org (M.D.); madycissoko@ymail.com (M.C.); coulibalyd@icermali.org (D.C.); 3Institut de Recherche pour le Développement (IRD), Institut National de la Santé et de la Recherche Médicale (INSERM), Aix Marseille Université (AMU), 13005 Marseille, France; 4Direction Générale de la Santé et Hygiène Publique, Sous-Direction Surveillance Épidémiologique, Bamako BP 233, Mali; berthe_enet@hotmail.com; 5Direction Régionale de la Santé de Tombouctou, Tombouctou BP 59, Mali

**Keywords:** incidence, malaria, Seasonal Malaria Chemoprevention, Mali

## Abstract

Context: In Mali, malaria transmission is seasonal, exposing children to high morbidity and mortality. A preventative strategy called Seasonal Malaria Chemoprevention (SMC) is being implemented, consisting of the distribution of drugs at monthly intervals for up to 4 months to children between 3 and 59 months of age during the period of the year when malaria is most prevalent. This study aimed to analyze the evolution of the incidence of malaria in the general population of the health districts of Kati, Kadiolo, Sikasso, Yorosso, and Tominian in the context of SMC implementation. Methods: This is a transversal study analyzing the routine malaria data and meteorological data of Nasa Giovanni from 2016 to 2018. General Additive Model (GAM) analysis was performed to investigate the relationship between malaria incidence and meteorological factors. Results: From 2016 to 2018, the evolution of the overall incidence in all the study districts was positively associated with the relative humidity, rainfall, and minimum temperature components. The average monthly incidence and the relative humidity varied according to the health district, and the average temperature and rainfall were similar. A decrease in incidence was observed in children under five years old in 2017 and 2018 compared to 2016. Conclusion: A decrease in the incidence of malaria was observed after the SMC rounds. SMC should be applied at optimal periods.

## 1. Introduction

More than half of the world’s population lives in areas at risk of malaria, considered to be the most frequent parasitic disease in humans. It remains one of the major challenges that health policies and systems face, although significant progress has been made [1]. In Mali, malaria is the primary reason for consultation during the season of high transmission. According to the National Health Information System (SNIS), in 2018, health facilities recorded 2,345,481 malaria cases [2]. Malaria was the pathology that caused more deaths in 2018, with 1178 deaths, according to the SNIS [2].

There are five climatic zones in Mali: the pre-Guinean zone, with an annual rainfall of more than 1100 mm; the Sudanian zone (900–1100 mm); the Sudano-Sahelian zone (500–900 mm); the Sahelian zone (250–500 mm), the desert zone, or Sahara, with an annual rainfall of less than 250 mm; and the inner delta of the Niger River.

The geographical location of Mali and its climatic conditions make malaria a major problem in almost the entire country. Regarding the fight against malaria, the country’s authorities have set up control strategies through the National Malaria Control Program (NMCP). These strategies involve the diagnosis and treatment of clinical cases, epidemiological surveillance, vector control with insecticide-treated nets and indoor residual spraying, and chemoprevention for target groups [3,4]. They have all been recommended by the World Health Organization (WHO) and implemented by the NMCP. Among these strategies, Seasonal Malaria Chemoprevention (SMC) for children from 3 to 59 months of age is one of the most important, effective, inexpensive, and safe strategies [5].

In studies on the effectiveness of control strategies in Mali in the health districts of Sadiola and Segou, the number of malaria cases was reduced by 70% after intrahousehold spraying campaigns [6,7]. For chemoprevention, the National Malaria Control Program (NMCP) recommends Seasonal Malaria Chemoprevention (SMC) with the administration of Sulfadoxine–Pyrimethamine and Amodiaquine (AQ-SP) in children and in pregnant women the use of intermittent preventative treatment with Sulfadoxine–Pyrimethamine (SP). SP should be given to pregnant women, three tablets every month, from the second trimester of pregnancy. SMC has been recommended by the WHO since 2012 in Sahelian countries, where more than 60% of clinical malaria cases occur within a maximum of four months [5,8,9].

This strategy consists of administering a single dose of SP and a dose of Amodiaquine for three days, in four rounds, to children aged 3–59 months and, recently, 5 to 10 years old [10,11]. SMC pilot studies showed encouraging results, with a reduction in the number of malaria and anemia cases in children aged 6–59 months. These results, based on an observational study, show a reduction in the risk of confirmed malaria by 40% [12,13]. Recent studies have produced results on the effectiveness of SMC in Mali in a health district in Ivory Coast and in Senegal [11,13,14,15].

After the administration of SMC, the authors of a study in Ouessebougou in central Mali measured the level of IgG-type antibodies against specific malaria antigens. These results show the presence of the antibodies [10]. However, the IgG amount was not proportional to the number of passages received by the child but was higher compared to children who did not receive SMC [16].

Despite the scaling up of these strategies, malaria continues to be prevalent and still causes morbidity and deaths, as shown by the statistics cited above. It is in this context that the National Malaria Control Program (NMCP) and the Malaria Research and Training Center (MRTC), in collaboration with the World Bank, initiated a study in Mali to monitor malaria incidence in the context of the scaling up of SMC. We conducted a secondary analysis of the data from this study in five health districts Kati, Sikasso, Kadiolo, Yorosso, and Tominian, located in different epidemiological settings where districts share borders with other countries and where the World Bank is supporting NMCP to implement SMC as their effort to support the Malian government in the fight against malaria. This malaria secondary data analysis is in part monitoring the malaria incidence trend using routine health system data in the context of scaling up SMC in order to guide the NMCP toward a targeted and adapted fight against malaria.

## 2. Method

The study protocol was approved by the Ethics Committee of the Faculty of Medicine and Odonto-Stomatology/Faculty of Pharmacy, University of Sciences, Techniques and Technologies of Bamako, N°. 2017__131__/CE/FMPOS, and all regulatory requirements were met. The anonymity of respondents and documents is preserved.

### 2.1. Study Location and Population

This study was carried in five health districts belonging to different epidemiological settings of malaria transmission and funded by the World Bank within the framework of the Neglected Tropical Disease (NTD) and Malaria project.

Kati health district: located about 15 km (Kati city) northwest of Bamako.

Kadiolo health district: located about 470 km (Kadiolo city) from Bamako, in the extreme south of Mali.

Sikasso health district: located at about 370 km (Sikasso city) from Bamako, in the extreme south of Mali.

Tominian health district: located about 464 km (Tominian city) from Bamako, north-east of the Segou region of Mali.

Yorosso health district: located about 426 km (Yorosso city) from Bamako. The climatic characteristics of the temperature and rainfall of the study sites are presented in Table 1.

According to the different climatic zones in Mali, the study sites are located in the Sudanian and Sahelian zones (Tominian and Kati, respectively) and in the South Guinean zone (Kadiolo, Yorosso and Sikasso) (Table 1) [17,18].

The Kati, Tominian, and Yorosso health districts have an annual rainfall between 500 and 800 mm, and the average temperature was 30 °C for Kati and Tominian. Those in southern Mali (Sikasso and Kadiolo) are more humid and rainfall can exceed 1000 mm per year.

### 2.2. Data and Sources

This is a transversal study analyzing routine malaria data from 5 health districts: Kati, Kadiolo, Sikasso, Yorosso, and Tominian. All malaria cases confirmed by the rapid diagnostic test for malaria (RDT), available for diagnosis in health centers, or a thick smear at the level of health facilities, reported through the monthly reports and recorded in the District Health Information Software (dhis2) version 2, were exhaustively collected. The databases of validated monthly malaria reports and the dhis2 software were used to extract monthly malaria data from the concerned health districts during the study period. The Nasa Giovanni site (https://giovanni.gsfc.nasa.gov/giovanni/) was used to download the meteorological data (mean, minimum, maximum temperature, relative humidity, cumulative rainfall) in the 5 health districts. All aggregated data were on a monthly time scale.

For the analysis of the relationship between meteorology and malaria incidence, we classified the health districts by the climatic zone in Mali. We selected two climatic zones: the Northern Sudan zone and Sahel for the first climatic zone and Sudan Guinean for the second climatic zone. Additional meteorological data were collected by the climatic zone on the Giovani site.

### 2.3. Data Analysis

For malaria incidence, we compared the annual incidence by the health district from 2016 to 2018. The percentage change in incidence between 2016 and 2018 in each of the 5 study sites was defined according to the WHO method:Rate variation = (Annual incidence (_2016_) − Annual incidence (_2018_))/Annual incidence (_2016_) * 100(1)

For the meteorological analysis, we performed a descriptive analysis of monthly malaria and meteorological (rainfall, temperature, relative humidity) time series data analysis from 2016 to 2018 with the graphs and tables.

The ANOVA test and the nonparametric Kruskal–Wallis test were used to compare the monthly averages calculated over the study period of rainfall accumulation, average relative humidity, and average temperature in the different health districts.

To determine the relationship between malaria and meteorological variables, we performed a Principal Component Analysis (PCA), which reduced the dimensions of the meteorological variables and took into account their collinearity [19]. A Generalized Additive Model (GAM) was used to study the relationship between malaria incidence and meteorological variables [20]. The negative binomial distribution was used to account for the overdispersion of the malaria case, which was the dependent variable during the study period [21]. The population of the study sites was transformed into the logarithmic form for obtaining the risk ratio of malaria incidence at each health district. The SMC variable was used among the covariates. The choice of this model was justified by the nonlinearity of malaria data, which is characterized by seasonality and trend [21]. The statistics were considered significant with a value of *p* ≤ 0.05.

Formule GAM:(2)$$log\left(Cases\left(T\right)\right)=log\left(Population\left(T\right)\right)+f_{1}\left(Dim1\right)+f_{2}\left(Dim2\right)+\varepsilon$$

### 2.4. Software and Packages

The various statistical analyses were performed using R software version 3.4 (R Development Core Team, R Foundation for Statistical Computing, Vienna, Austria) packages {mgcv}, {FactoMineR}, and {Factoextra}. Paint.net version 4.2.13 (Warren Paint and Color C., Nashville, TN, USA) was used for image processing.

## 3. Results

### 3.1. Evolution of Malaria Incidence in the Five Health Districts

Monthly malaria incidence from 2016 to 2018 showed an increase from July to November in all study health districts corresponding to the rainy season period. There was a decrease in malaria incidence from 2016 to 2018, except in the Yorosso and Sikasso health districts (Appendix A). From January to June, there was a decrease in malaria incidence in the general population, which confirms the seasonality of malaria. The high transmission season is longer (five to six months), varying from June to November in the health districts of Kadiolo, Yorosso, and Sikasso, located in the Sudano-Guinean facies, and from four to five months in the health districts of Kati and Tominian, located in the northern Sudanian and Sahelian profile (Figure 1 and Figure 2).

The time series of monthly malaria incidence in the five health districts show a peak in the rainy season every year. This period of high transmission is from July to December or even January of the following year. Afterward, a decrease in incidence is observed.

The health districts of Kadiolo, Yorosso, and Tominian had the highest average monthly aggregate incidence, which ranged from 160 to 290 cases per 10,000 persons per month. The lowest incidences were observed in the health districts of Sikasso and Kati (Figure 1).

The health districts of Kadiolo (median of 200 cases per 10,000 persons) and Yorosso (median of 210 cases per 10,000 persons-month) had the highest incidence rate in October. The health district of Sikasso (median of 50 cases per 10,000 persons-month) had the lowest monthly incidence rate.

### 3.2. Estimated Monthly Incidence in Under-Fives and Five or Over Five Years Olds and SMC

In all five districts, the monthly incidence of under-fives is higher than the monthly incidence of over-fives from 2016 to 2018. The highest incidences were recorded during the rainy season and the lowest during the dry season. Peaks for both groups are generally obtained between July and October. SMC has been regularly implemented in all the health districts from August to November in 2016 and from July to October from 2017 to 2018. However, a decrease in incidence was only noted after the third and fourth rounds. The most important rainfall was obtained from July to August over the three years, which corresponds to the period of SMC. Monthly incidence peaks are observed at the end of the rainy season in September and October (Appendix A). The figures show that Seasonal Malaria Chemoprevention (SMC) was always implemented after the start of the high transmission period in all the health districts.

The percentage change in incidence between 2016 and 2018 at the five sites was defined according to the WHO method.

### 3.3. Analysis of the Meteorological Variables into the Five Health Districts

#### 3.3.1. Univariate Analysis of Overall Meteorological Variables between the Health Districts

Using the meteorological data collected in each district, we assessed their relation with malaria incidence.

#### 3.3.2. Miltivariiate Analysis of Overall Malaria Incidence and Meteorological Variables

The Principal Component Analysis (PCA) reduced the size of the meteorological variables in two dimensions (Dim) that explain the inertia to more than 80%. Dim-1 consisted of relative humidity and inversely average and maximum temperatures. Dim-2 consisted essentially of the minimum temperature and rainfall from the same climatic zone (Figure 3). The contribution of rainfall (40%) is higher than the minimum temperature (30%) in the Sahel zone (Figure 3a) and inversely in the Sudano-Guinean zone in Dim-2 (Figure 3b).

The Generalized Additive Model (GAM) was used to model the overall impact according to the components derived from the PCA by climatic zone. The different meteorological variables synergized using PCA made it possible to look for the relation between the meteorological variables and malaria incidence in the two climatic zones of this study. The Dim-1 component was composed positively of relative humidity and negatively of average maximum temperature, and the Dim-2 component was composed of rainfall and minimum temperature. In the Northern Sudan and Sahel zone, Dim-1 consisted of relative humidity and inversely average and maximum temperatures, and Dim-2 consisted essentially of the minimum temperature and rainfall and was significantly associated with overall malaria incidence (*p* > 0.001) in multivariate analysis with a linear relationship of Dim-2 (Table 4). The explained deviance was 74%.

In the Sudano-Guinean zone, Dim-1 consisted of positive relative humidity and inversely average maximum temperatures, and Dim-2 consisted essentially of the minimum temperature and rainfall were significantly associated with overall malaria incidence (*p* < 0.001) in the multivariate analysis (Table 4). The explained deviance was 50%.

## 4. Discussion

The limitations of our study were the missing lag between the overall incidence and weather components because weekly data were not available. The specific coverage rates of the control strategies (possession and the use of LLINs, coverage of SMC) could help explain some of the differences in the variation in incidence in the health districts, although they generally confirmed that the usage of LLINs is expected similar at those districts.

The objective of this work was to analyze the evolution of malaria incidence in the general population of the study districts from routine data and determine the relationship between malaria incidence and environmental variables such as temperature, relative humidity, and rainfall in the five health districts (Kati, Kadiolo, Sikasso, Yorosso, and Tominian) as part of the evaluation of the impact of Seasonal Malaria Chemoprevention from 2016 to 2018.

The evolution of the monthly incidence of malaria shows that the period of high malaria transmission in Mali coincides with the rainy season and the formation of water reservoirs and ponds. This creates a favorable environment for the development and multiplication of mosquitoes responsible for malaria transmission, which could explain the increasing evolution of malaria incidence in the five health districts with significant peaks, especially between July and October.

Incidence decreased between 2017 and 2018 compared to 2016 except in the Yorosso district (Appendix A and Table 2). This could be explained by the door-to-door strategy adopted by the NMCP started in 2017, which is supposed to have better SMC coverage compared to the fixed strategy performed previously.

The Kadiolo health district generally presented the highest incidences, followed by Tominian, Yorosso, Kati, and Sikasso. The duration of peak malaria transmission differs from one district to another, confirming the ecological impact on malaria. Several studies carried out in Mali and around the world show that the incidence of malaria is higher during the rainy season, notably the study conducted by Dolo et al. in 2003 in Bancoumana, Mali [22] and the study conducted by Sissoko et al. in a suburban area along the Niger River in Mali [23] Sotuba and Cissoko et al. in Dire in the Malian Sahel area [6] in the Peruvian Amazon region during a 2016 study found that malaria incidence was the highest between February and July, which coincides with the rainy season in Peru. The incidence was lower in urban areas than in rural areas. There was relatively less malaria in Sikasso city, an urban area, which would be less favorable to the proliferation of Anopheles, the malaria vector, followed by Kati, a less humid area. In the literature review, it was seen that urbanization was unfavorable to the development of Anopheles, which is contrary to the results of Sissoko et al., who, in their 2017 study in Sotuba, reported that despite increasing urbanization, no reduction in malaria incidence was observed [23].

The incidence of malaria in under-fives based on meteorological variables and SMC was higher than the incidence of over-fives between 2016 and 2018 in all districts. This may explain the vulnerability of children to malaria [24]. In children under five years of age, it is believed that the immune system is not mature enough to cope with malaria parasites. It is also possible that, according to consultation data from health facilities, children under five years of age are more likely to have attended health centers than those over five years of age. Some free care such as free malaria treatment is an argument in favor of higher consultation in health centers for this age group.

The peaks for both groups were obtained between July and October, with the largest peak in September 2016 for children under five years of age compared to 2017 and 2018. This result could be explained by the door-to-door strategy adopted by the NMCP starting in 2017. This strategy provides better coverage of SMC compared to the fixed strategy. The period covered by SMC was from July to October (Figure 2), hence the extreme evolution of malaria incidence (peaks) obtained during this period despite the four cycles of SMC, which are supposed to decrease the incidence of malaria in children under five years old. This could be explained by the fact that during SMC, malaria is detected in the community and not in the health facilities.

However, a decrease in incidence was noted in children under five years of age just after the third or fourth round of SMC, particularly between October and November. This could be due to the positive impact of SMC on the improvement of health indicators, particularly the incidence of malaria among children under five years of age.

Maximum temperatures were obtained during the month of April, going up to 40 °C. There was an upward shift in the temperature curve from January to June, and the rate of malaria incidence decreased in the population. This can be explained by the fact that extreme temperatures (above 35 °C) reduce mosquito longevity as humidity decreases [25].

Correlation analysis between incidence and the time variable (months) allowed research on the impact of control strategies, including SMC in children under five years old. The Kadiolo health district had a strong difference with 24.14% in malaria incidence reduction in 2018 compared to 2016, and the Yorosso health district had a weak negative slight difference.

The heaviest rainfall was obtained from July to August over the three-year period, which corresponds to the period of rainfall and the start of SMC. During this period, the incidence of malaria increased, and the highest malaria incidence was obtained at the end of the rainy season (September–October). Rainfall is responsible for the creation of mosquito breeding sites. The figures show that Seasonal Malaria Chemoprevention (SMC) was always implemented after the start of the high transmission period in all the health districts. This delay could be related to the availability of funding and the heterogeneity of malaria transmission. With climatic variations, rainfall can start earlier [26,27]. This observation needs to be confirmed by future work (Appendix A).

The calculation of the percentage variation in incidence between 2016 and 2018 in the five health districts shows a decrease in malaria incidence at some sites. This decrease can be attributed to the implementation of control strategies but also to seasonal variations linked to the climate. However, we observed an increase in the number of cases in the health districts of Sikasso and Yorosso (Table 2). This increase could be explained by the increase in rainfall in 2018 and extension until November (Appendix A) [28,29].

### The Comparison between Meteorological Variables in the Health Districts

The mean monthly relative humidity was significantly different (*p* < 0.001) across the study districts over the three years. In contrast, the monthly averages of the mean temperature and mean rainfall were not significantly different across the study districts during the three years (Table 3).

This study reveals that relative humidity had a greater influence on malaria transmission rainfall, as described elsewhere [30,31]. If the relative humidity is different between the study sites, then the transmission of malaria will also be different.

Several studies have reported that heavy rainfall is associated with malaria incidence [32]. The association between relative humidity and malaria reflects the indirect role of rainfall and temperature on density, survival, longevity, ineffectiveness, and vector capacity in malaria transmission.

The combination of meteorological variables allowed the identification of two main components, the first of which was Dim-1, consisting of relative humidity (average of minimum and maximum) positively and negatively the average and maximum temperature. The second dimension, called Dim-2, consisted of minimum temperature and rainfall. Dim-1 was significantly correlated with malaria incidence, indicating a significant increase in malaria incidence (*p* < 0.001). Dim-2 was significant (*p* < 0.05) in the Sudano-Guinean zone (Table 4).

The main meteorological variable associated with malaria is a combination of humidity, rainfall, and temperature (mean and minimum), described by several previous studies [23]. This linear correlation has been produced without lag. In other regions, however, the time lag between rainfall and malaria cases might be different. Rainfall is not the only factor related to malaria transmission. Indeed, the humidity, temperature, land cover, and land use are also important factors [18,23,24]. As malaria transmission is known to be heterogeneous, depending on the geographical and environmental context, the difference in malaria transmission between different areas is not surprising.

Most studies in the literature have shown that rainfall and humidity increase the risk of malaria by developing suitable breeding sites and increasing mosquito density [30,33,34]. Studies show the persistence of malaria transmission despite the intensification of control strategies. If we take into account the role of climate, strategies must also be adapted over time, targets, and in different eco-climatic zones [3,9,35,36]. At the start of the rainfall season in Mali, the temperature tends to lower, relative humidity and vegetation increase, and larval breeding sites form near the houses [37,38,39]. The hibernating mosquitoes return to life and lay eggs [40]. As the eco-climatic conditions are favorable, the length of the breeding cycle decreases from five weeks to one week, depending on the species. The vegetation offers protection for the eggs, nymphs, and larvae [40]. The mosquito population multiplies exponentially. Patients and asymptomatic carriers of the parasite, which are human reservoirs, allow the transmission to continue [41]. After the rainfall season, the temperature gradually increases. This causes low relative humidity and vegetation cover. Mosquitoes migrate from relatively humid areas with little water to maintain their survival [42,43].

This climate is the main factor that modulates malaria transmission in Mali. This period should be targeted for implementing control strategies (SMC, impregnated mosquito nets, and indoor residual spraying). However, it must be specific to each area as the rainy season varies from one locality to another.

## 5. Conclusions

We see that the dynamics of malaria transmission, seasonality, and variability remain intact despite the context of SMC and other strategies. However, the incidence of malaria decreases after the SMC cycles, and the incidence of malaria relatively decreased in the under-five age group in 2017 and 2018 compared to 2016. The results of this study show that meteorological factors have a definite impact on malaria incidence.

This study shows that preventative control strategies such as SMC can have a significant impact if optimal start-up periods are defined in the health districts.

## Figures and Tables

**Figure 1 ijerph-18-00840-f001:**
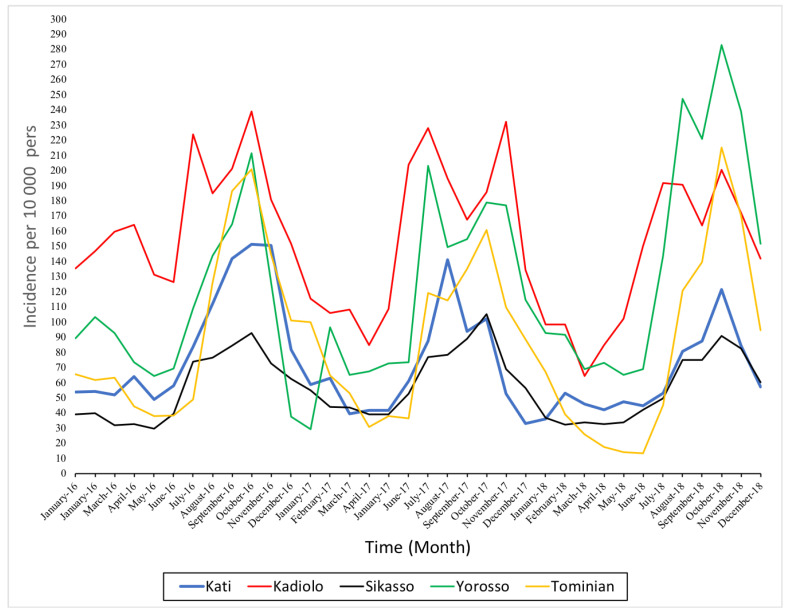
Monthly malaria incidence by health district from 2016 to 2018.

**Figure 2 ijerph-18-00840-f002:**
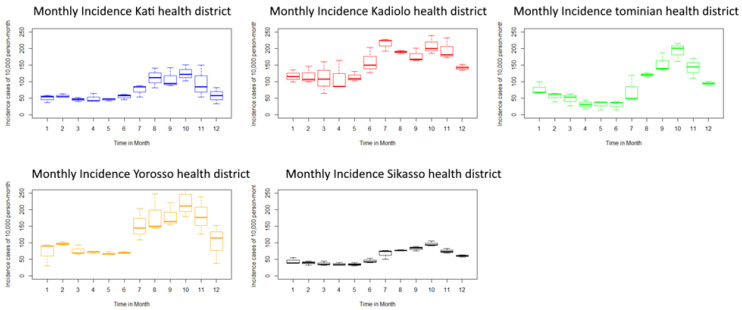
Average monthly malaria incidence by health district from 2016 to 2018 for 10,000 pers.

**Figure 3 ijerph-18-00840-f003:**
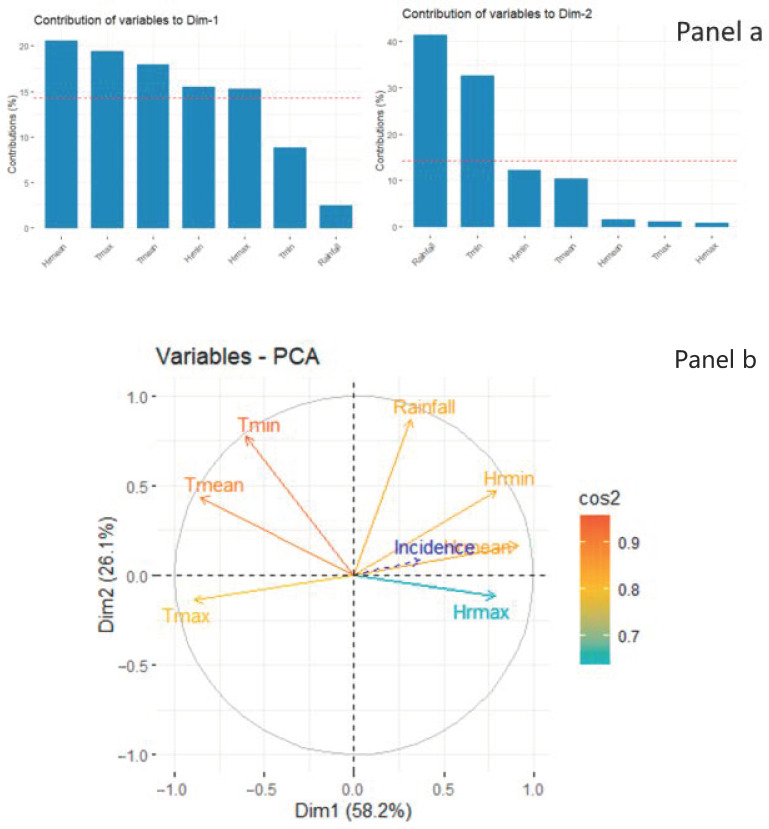
Graph of PCA with the contribution of meteorological variables: (**a**) Sahel zone; (**b**) Sudano-Guinean zone.

**Table 1 ijerph-18-00840-t001:** Climatic characteristics (rainfall and temperature) at the study sites from 2016 to 2018.

Health District	Average Annual Rainfall (mm)	Average Annual Temperature (°C)
Kati	789.6	30
Kadiolo	1000	27.6
Sikasso	1100	27
Tominian	500	30
Yorosso	681.7	28

**Table 2 ijerph-18-00840-t002:** Percentage change in incidence between 2016 and 2018.

Health District	Sum of Annual Cases			Rate Variation (%)
	2016	2017	2018	
Kati	66,781	53,276	50,660	24.14
Kadiolo	62,314	58,614	53,522	14.11
Sikasso	41,720	47,620	42,206	−1.16
Tominian	31,096	30,033	28,325	8.91
Yorosso	34,511	38,274	49,692	−43.99

Percentage change shows a reduction in malaria incidence in the three health districts (Kati, Kadiolo and Tominian) between 2016 and 2018.

**Table 3 ijerph-18-00840-t003:** Comparison of meteorological variables between the five health districts.

Variables	*p*-Value	Test
Average monthly rainfall	0.296	ANOVA
Average monthly temperature	0.686	ANOVA
Average monthly relative humidity	<0.001	Kruskal–Wallis

Univariate analysis of the different health districts showed a significant difference in the relative humidity (Kruskal–Wallis test, *p* < 0.001). However, there was no significant difference in mean rainfall (ANOVA test, *p* = 0.296) and temperature (ANOVA test, *p* = 0.696) among the five health districts (Table 3).

**Table 4 ijerph-18-00840-t004:** Multivariate analysis between the meteorological axis and malaria incidence with General Additive Model (GAM).

Nonparametric Part	*p*-Value	Deviance
Dim-1 (Northern Sudan and Sahel Zone)	<0.001	
Dim-2 (Northern Sudan and Sahel Zone)	<0.001	74%
Dim-1 (Sudano-Guinean)	<0.001	
Dim-2 (Sudano-Guinean)	0.006	50%

## Data Availability

Data can be shared as needed.

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
