# Peer review of "Evolution of Malaria Incidence in Five Health Districts, in the Context of the Scaling up of Seasonal Malaria Chemoprevention, 2016 to 2018, in Mali"

_ijerph, 2021, doi:10.3390/ijerph18020840_

Round 1

Reviewer 1 Report

Comments:

  1. English of the manuscript needs to be improved considerably with spelling checks.
  2. The introduction is not complete. I would recommend adding a paragraph discussing the idea behind the study.
  3. Please give a detailed introduction of SMC, add the earlier literatures if present and in case it is a novel study then explain why do you think this is a good idea? The number of articles cited/referred throughout the manuscript looks very less. i would recommend adding more citations. 
  4. Section 2.1- Rephrase- Follow same pattern of either giving range or average value. Add temperature for all the locations. I would recommend adding a table to give a clear comparison of all these values.
  5. Figure 1- Add year along with month to show the separation between 2006-08.
  6. Figure 2- The error bars does not show significant differences between districts as written by the author in the text? can you elaborate?
  7. Figure 2- Add a figure showing difference between July to December to show the differences between districts.
  8. I recommend doing a correlation analysis between different factors to show their relationship.
  9. Add figure for PCA analysis for the best axes.
  10. Supplementary Figure- What does the X axis represents? Add in all supplementary figures.

Author Response

Response to Reviewer 1 Comments

Response: We thank you for agreeing to review this study. We have taken all your comments into account in revising our manuscript.

You can see all the changes in the manuscript in track changes mode and our responses below in each of your query in bold.

English of the manuscript needs to be improved considerably with spelling checks.

Response: Please note that the manuscript has been improved with spelling checks

  1. The introduction is not complete. I would recommend adding a paragraph discussing the idea behind the study.

Response:  We made improvements to the introduction in discussing more the idea behind the study.

  1. Please give a detailed introduction of SMC, add the earlier literatures if present and in case it is a novel study then explain why do you think this is a good idea? The number of articles cited/referred throughout the manuscript looks very less. i would recommend adding more citations. 

Response: We have added a paragraph to define SMC and explain the strategy. We have added additional references as well (See lines 42-87)

  1. Section 2.1- Rephrase- Follow same pattern of either giving range or average value. Add temperature for all the locations. I would recommend adding a table to give a clear comparison of all these values.

Response: We have provided the same pattern and provided more description (See Lines 99:121) and inserted a synthetic table providing more data including temperature for all locations and made a comparison among districts as well (see Table 1).

  1. Figure 1- Add year along with month to show the separation between 2006-08.

Response: This figure has been edited to add the month and year (See Figure 1).

  1. Figure 2- The error bars does not show significant differences between districts as written by the author in the text? can you elaborate ?

Responses: 

To see the changes in impact between 2016 and 2018, we have calculated the percentage change between 2016 and 2018 and update findings throughout the manuscript (See Table 2).

Spearman's correlation analysis was carried out on the mean annual incidence and the time variable from 2016 to 2018 (See line 214-219).

  1. Figure 2- Add a figure showing difference between July to December to show the differences between districts.

Response: We have taken your recommendation into account by adding a graph (Boxplot) showing the monthly malaria incidence from 2016 to 2018 for each district (See Figure 2).

I recommend doing a correlation analysis between different factors to show their relationship.

Response: We have taken your recommendation into account and did the correlation analysis (See line 214-219) as well as performed the mean variation of malaria incidence and meteorological parameters among districts (See Table 3).

  1. Add figure for PCA analysis for the best axes.

Response: We have added figures for the PCA analysis (Figure 3)

  1. Supplementary Figure- What does the X axis represents? Add in all supplementary figures.

Response: The x-axis represents the months of 2016-2018. We have modified the graphs to make reading easier (See the Supplementary Figures).

Reviewer 2 Report

accepted with major changes

This work has two objectives, one to measure a prevalence after an intervention or programs

they conclude that they may be associated but do not report anything about the program. The first should be clearer, because it assumes that I improve the incidence of malaria due to the interventions, but that cannot be concluded, it is difficult.

climatic analysis climate analysis makes it interesting and improves work

 It is difficult to associate them but if I affirm that it could have an impact, it is good to know or evaluate or have more of the program. It is not clear if it is only in children or why they give relevance to children

 The other objective is to associate malaria with the climate. let's say gave the article a plus. but they do not say well how this was done methodologically, they do not explain the model well

Author Response

Response to Reviewer 2 Comments

Response: We thank you for agreeing to review this study. We have taken all your comments into account in revising our manuscript.

You can see all the changes in the manuscript in track changes mode and our responses below in each of your query in bold.

This work has two objectives, one to measure a prevalence after an intervention or programs

they conclude that they may be associated but do not report anything about the program. The first should be clearer, because it assumes that I improve the incidence of malaria due to the interventions, but that cannot be concluded, it is difficult.

climatic analysis climate analysis makes it interesting and improves work

 It is difficult to associate them but if I affirm that it could have an impact, it is good to know or evaluate or have more of the program. It is not clear if it is only in children or why they give relevance to children

Responses: With regard to the first point, SMS is a proven prevention strategy in malaria endemic Sahelian countries for children from 3 to 59 months. We clarify this into the introduction and discussion section.

We actually wanted to assess the evolution of the malaria incidence over time based on routine data from the health structures. The incidence has decreased at some points (See supplementary graph). This decrease can be attributed to SMC and other control strategies. However, as malaria is a climate-dependent disease, this decrease could be due to seasonal variation. This result has been discussed with other authors (See the Discussion section).

The peaks for both age groups were obtained between July and October, with the largest peak in September 2016 for children under 5 years of age compared to 2017 and 2018. This result could be explained by the door-to-door strategy adopted by the NMCP starting from 2017. This strategy provides better coverage of the SMC compared to the fixed strategy. The period covered by the SMC was from July to October hence the extreme evolution of malaria incidence (peaks) is obtained during this period and this despite the 4 cycles of the SMC which is supposed to decrease the incidence of malaria in children under 5 years old.  This could be explained by the fact that during the SMC, malaria is detected in the community and not in the health facilities.

However, a decrease in incidence was noted in children under 5 years of age just after the third or fourth round of the SMC, particularly between October and November. This could be due to the positive impact of the SMC on the improvement of health indicators, particularly the incidence of malaria among children under 5 years of age (See the Discussion section).

We have added also the correlation analysis of incidence and the time variable (2016-2018) and the meteorological variables among the districts to assess the relationship of those variables

To see the changes or impact, we have calculated the percentage change between 2016 and 2018 and update findings throughout the manuscript (See Table 2).

The model of multivariate analysis is now better explained into the methodology section.

The other objective is to associate malaria with the climate. let's say gave the article a plus. but they do not say well how this was done methodologically, they do not explain the model well

Response: For the method used, we have taken into account your recommendations in providing more details in the methodological section and the result section in explaining the model and added the references used.

Reviewer 3 Report

In this manuscript, Sacko et al. describe the evolution of malaria incidence in 5 health districts of Mali from 2016 to 2018. They show the relationship between malaria incidence and meteorological factors, including relative humidity, rain fall and temperature. The malaria incidence of under 5 years old children age group decreased after Seasonal Malaria Chemoprevention (SMC). Overall, this is a fairly straightforward study provides insight into the prevalent malaria in Mali. I have listed some suggestions which could improve the manuscript.

  1. Line 86: All malaria cases confirmed by the rapid diagnostic test (RDT), how did the test conducted? Please clarify.
  2. Line 90: Add the website address of Nasa-Giovanni site.
  3. Line 102: “Anova” should be “ANOVA” and “Kruskall-wallis” should be “Kruskall-Wallis”.
  4. Line 108: “Painet. net” should be “Paint. net”.
  5. Line 114: There are no Figure 3 to 7, it should be Figures S1-S5.
  6. Lines 113-114: I can’t find a significant decrease of malaria incidence (2017 and 2018) compared to 2016 in Yorosso district in Fig S5 or Fig 1. Please clarify.
  7. Figure 1: Please add a line under x axis to mark the year 2016 (January to December), 2017 (Jan to Dec), 2018 (Jan to Dec).
  8. Lines 126-127: I was interested more the reason of lowest malaria incidences in Sikasso and Kati. The chemoprevention is better than other districts? Can you discuss more?
  9. Line 124: “(Figures S1, S2, S3, S4, S5)” should be “(Figures S1-S5)”.
  10. Table 2: “Anova” should be “ANOVA” and “Kruskall-wallis” should be “Kruskall-Wallis”.
  11. Line 168: “.” before the word “which” should be “,”.
  12. Line 173: “5 Health Districts” should be “five health districts”.
  13. Lines 174-175: Can you provide more information about SMC? Insecticide spay or pyrethroids treated net?
  14. Lines 181-182: See question 6, I can’t find a significant decrease and the Figure number is not correct.
  15. Line 228: “(Figures S1, S2, S3, S4, S5)” should be “(Figures S1-S5)”.
  16. Line 240: “Temperature” should be “temperature”.
  17. Line 241: “[8,13,13,14]” should be “[8,13,14]”.
  18. Line 248: “(p <0.000)” should be “(p <0.0001)”.
  19. Lines 267, 270, 276, 279: Supplementary figures should be named as “Figure Sx”, such as “Figure 1” should be “Figure S1”, “Figure 2” should be “Figure S2”, “Figure 4” should be “Figure S4”, “Figure 5” should be “Figure S5”.
  20. Lines 244-245: Can you discuss more about the rainfall, humidity and the risk of malaria?
  21. It would be useful to discuss in more detail the role of malaria mosquito Anopheles in the malaria incidence and mortality, the relationship of mosquito population and meteorological factors in Mali. A discussion of mosquito population changes from dry season to wet season should be added, including the mosquito larval density in the pool/wetland from five health districts. More references about malaria mosquito control should be cited.
  22. For the data throughout the manuscript, when you indicated “decrease”, “different”, you must have statistical analysis to confirm significant (p value?). I can’t see a decrease/increase by eyes.

Author Response

Response to Reviewer 3 Comments

English of the manuscript needs to be improved considerably with spelling checks.

Response: We thank you for agreeing to review this study. We have taken all your comments into account in revising our manuscript.

You can see all the changes in the manuscript in track changes mode and our responses below in each of your query in bold.

In this manuscript, Sacko et al. describe the evolution of malaria incidence in 5 health districts of Mali from 2016 to 2018. They show the relationship between malaria incidence and meteorological factors, including relative humidity, rain fall and temperature. The malaria incidence of under 5 years old children age group decreased after Seasonal Malaria Chemoprevention (SMC). Overall, this is a fairly straightforward study provides insight into the prevalent malaria in Mali. I have listed some suggestions which could improve the manuscript.

Response: We thank you for agreeing to review this study. We have taken all your comments into account in revising the manuscript.

  1. Line 86: All malaria cases confirmed by the rapid diagnostic test (RDT), how did the test conducted? Please clarify.

Response: All malaria cases confirmed by the rapid diagnostic test for malaria (RDT) available for diagnosis in health facilities or thick smear reported through the monthly reports and recorded in the District Health Information Software (DHSI2) version 2 were exhaustively collected. This has been clarified into the method section (See line 130-131)

  1. Line 90: Add the website address of Nasa-Giovanni site.

Response: We have added the address of the Giovani site where meteorological data can downloaded (see line 135)

  1. Line 102: “Anova” should be “ANOVA” and “Kruskall-wallis” should be “Kruskall-Wallis”.

Response: We have taken the observations into account (see line 151, and Table 3.).

  1. Line 108: “Painet. net” should be “Paint. net”.

Response: The correction was made in the manuscript (see line 172).

  1. Line 114: There are no Figure 3 to 7, it should be Figures S1-S5.

Response: The correction was made in the manuscript (see Figures S1-S5 and Table 1).

  1. Lines 113-114: I can’t find a significant decrease of malaria incidence (2017 and 2018) compared to 2016 in Yorosso district in Fig S5 or Fig 1. Please clarify.

Responses:

A clarification has been made throughout the manuscript related to the variation of malaria incidence including the decreasing aspect. There was no decrease noted in Yorosso district in 2016 compared to 2017 or 2018. We have made a table of the rate of variance of malaria incidence between 2016 and 2018 to support this assessment (See Table 2).

Spearman's correlation analysis was carried out on the mean annual incidence and the time variable from 2016 to 2018 (See line 214-219).

We have added a paragraph to the discussion on this matter as well. 

  1. Lines 126-127: I was interested more the reason of lowest malaria incidences in Sikasso and Kati. The chemoprevention is better than other districts? Can you discuss more?

Response: The lowest malaria incidences in Sikasso and Kati could be explained by the urbanization level of those districts compared to the three other districts. We have clarified this on the discussion section (See lines 313-318)

  1. Line 124: “(Figures S1, S2, S3, S4, S5)” should be “Figures S1-S5)”.

Response: We have made the correction to be “Figures S1-S5” (see line 179).

  1. Table 2: “Anova” should be “ANOVA” and “Kruskall-wallis” should be “Kruskall-Wallis”.

Response: We agree and have made the correction (See Table 3 and lines 235-237).

Line 168: “.” before the word “which” should be “,”.

Response: We had made the correction (see line 280).

  1. Line 173: “5 Health Districts” should be “five health districts”.

Response: We had made the correction (See line 294).

  1. Lines 174-175: Can you provide more information about SMC? Insecticide spay or pyrethroids treated net?

Response: We have provided more information into the introduction section about malaria control strategies in Mali including SMC, Insecticide spay or pyrethroids treated net (See lines 47:87).

  1. Lines 181-182: See question 6, I can’t find a significant decrease and the Figure number is not correct.

Responses: 

The Figure number error has been corrected.

To see the changes or impact, we have calculated the percentage change between 2016 and 2018 and updated the findings throughout the manuscript (See Table 2).

Spearman's correlation analysis was carried out on the mean annual incidence and the time variable from 2016 to 2018 (See line 214-219).

  1. Line 228: “(Figures S1, S2, S3, S4, S5)” should be “(Figures S1-S5)”.

Response: The correction was made into the manuscript to be Figures S1-S5.

  1. Line 240: “Temperature” should be “temperature”.

Response: It has been now changed to be “temperature” on that line.

  1. Line 241: “[8,13,13,14]” should be “[8,13,14]”.

Response: We have made the correction but the reference number has been edited.

  1. Line 248: “(p <0.000)” should be “(p <0.0001)”.

Response: We have made the correction on the indicated line.

  1. Lines 267, 270, 276, 279: Supplementary figures should be named as “Figure Sx”, such as “Figure 1” should be “Figure S1”, “Figure 2” should be “Figure S2”, “Figure 4” should be “Figure S4”, “Figure 5” should be “Figure S5”.

Response: We had made the correction in supplementary material as from Figure S1 to Figure S5.

  1. Lines 244-245: Can you discuss more about the rainfall, humidity and the risk of malaria?

Response: We have added a discussion paragraph on malaria risk and meteorological variables (See lines 360-412)

  1. It would be useful to discuss in more detail the role of malaria mosquito Anopheles in the malaria incidence and mortality, the relationship of mosquito population and meteorological factors in Mali. A discussion of mosquito population changes from dry season to wet season should be added, including the mosquito larval density in the pool/wetland from five health districts. More references about malaria mosquito control should be cited.

Response:  We have incorporated your suggestion into the discussion section (See lines 360-412)

  1. For the data throughout the manuscript, when you indicated “decrease”, “different”, you must have statistical analysis to confirm significant (p value?). I can’t see a decrease/increase by eyes.

Response: Your comment has been taken into account (See Table 2 and Table 3), and this point has been discussed (See lines 360:412).  

Round 2

Reviewer 1 Report

Thank you for accepting the comments and working on it. The manuscript looks much better in shape and I reccomend it for publication.

Author Response

Thank you for your review.

Regards

Reviewer 2 Report

Although many changes were made that improved the article, the confusion between the two objectives continues.

The description of the areas is very long. You do not need to describe the environmental part of the area as much if this is in the table.

The two methods: incidence and meteorological, are not clearly separated. In some moments it gets confused. In this case, I advise to explain the incidence first and then the meteorological separately. Place an item in the methodology for the study of the incidence and another for the meteorological.

The results must be presented in order first incidence and then meteorological

Author Response

Response to Reviewer 2 Comments

Response: We thank you for agreeing to review this manuscript for a second time. We have taken all your comments into account in revising it.

You can see all the changes in the manuscript in track changes mode and our responses below in each of your query in bold.

Although many changes were made that improved the article, the confusion between the two objectives continues.

We have reviewed the article part which making the confusion between the two objectives. The incidence of malaria has now been removed. We retained only the meteorological variables for the univariate tests (See method part: line 151, result part: lines 223-232 and discussion section: lines 360-366).

The description of the areas is very long. You do not need to describe the environmental part of the area as much if this is in the table.

We have shorten the description of the study sites (See lines 97-116)

The two methods: incidence and meteorological, are not clearly separated. In some moments it gets confused. In this case, I advise to explain the incidence first and then the meteorological separately. Place an item in the methodology for the study of the incidence and another for the meteorological.

We have now revised the manuscript to provide the incidence first and then the meteorological separately in the methods section: lines 142-152.

We have placed an item in the methodology for the study of the incidence and another for the meteorological (see the Data analysis in the method section: lines 143 & 150).

The results must be presented in order first incidence and then meteorological

We have updated the manuscript on the results section in presenting the malaria incidence first and then the meteorological factors (see the results section: lines 226-229 including Table 3 tile and line 251).

Reviewer 3 Report

The revised version looks good.

Author Response

Response to Reviewer 3 Comments

English of the manuscript needs to be improved considerably with spelling checks.

We thank you for agreeing to review this study again. We have taken your suggestions into account in reviewing the manuscript.

The manuscript has been reviewed by a professional to improve the English language.